# What is good grief support? Exploring the actors and actions in social support after traumatic grief

**Joanne Cacciatore**[1]*, **Kara Thieleman**[2], **Ruth Fretts**[3], **Lori Barnes Jackson**[4]

**1** School of Social Work, Associate Professor, Arizona State University, Phoenix, AZ, United States of America, **2** School of Social Work, Adjunct Faculty, Arizona State University, Phoenix, AZ, United States of America, **3** Atrius Health, Medical Doctor, Boston, MA, United States of America, **4** School of Social Work, Graduate Student, Arizona State University, Phoenix, AZ, United States of America

\* Joanne.Cacciatore@asu.edu

## Abstract

Social support seems to enhance wellbeing and health in many populations. Conversely, poor social support and loneliness are a social determinant of poor health outcomes and can adversely affect physical, emotional, and mental well-being. Social support is especially important in traumatic grief. However, the ways in which grieving individuals interpret and define social support is not well understood, and little is known about what specific behaviours are perceived as helpful. Using qualitative description and content analysis, this study assessed bereaved individuals' satisfaction of social support in traumatic grief, using four categories of social support as a framework. Findings suggest inadequate satisfaction from professional, familial, and community support. Pets emerged with the most satisfactory ratings. Further, findings suggest that emotional support is the most desired type of support following traumatic loss. Implications for supporting bereaved individuals within and beyond the context of the COVID-19 pandemic are discussed.

## Introduction

The COVID-19 pandemic has resulted in untimely deaths and social isolation for many people throughout the world, with some researchers expressing a concern for increased prevalence in prolonged grief disorder [1]. In addition, the culture-specific rituals around deaths have been disrupted, with many unable to grieve with family and friends [2]. Eisma et al. [1] call for improvements to care because bereavement, particularly when prolonged and unattended, is a serious concern for public health, and adverse mental and physical health outcomes have been well documented in the literature even prior to the pandemic [3, 4].

Social support has been studied as a means to enhance human health and well-being, and can be provided at low cost within medical, mental health, and community systems. Beginning in the 1980s, it was recognized that the lack of meaningful social relationships (or lack of meaningful relatedness) is a social determinant of poor health outcomes, associated with an increased risk of death from a variety of causes [5]. It is now more widely accepted that poor social support and loneliness can adversely affect physical, emotional, and mental well-being

**Data Availability Statement:** Anonymized data are included.

**Funding:** The author(s) received no specific funding for this work.

**Competing interests:** The authors have declared that no competing interests exist.

[6–8], is linked to an increased premature mortality risk [9, 10] and correlates to adverse health outcomes such as depression, psychosocial maladjustment, poor coping behavior, low health promotion behavior, compromised well-being, reduced quality of life, and self-actualization [11]. Loneliness, specifically, has been called "the 21st century social determinant" of poor outcomes in health.

There is also an inverse benefit to feeling supported and connected, particularly when faced with stressful life challenges. Social support improves mental, physical, and emotional health outcomes and has been studied extensively within varying disciplines over the last 40 years with similar results [12, 13]. Strong social support safeguards against the negative psychological and physiological responses to stress; it is a buffer of protection that aids coping. For example, social support improves cardiovascular, endocrine, and immune system health as measured by blood pressure, heart rate, epinephrine, norepinephrine, and cortisol levels and immune cells and antibodies [14].

Research on bereavement suggests that loneliness and inadequate social support are common, giving rise to concerns about the risks to emotional, mental, and physical health for grievers [4, 15]. For instance, some scholars have found a relationship between loneliness and post-bereavement depressive symptoms, adding to the global burden of mental illness [16]. This situation has been worsened considerably by the COVID-19 pandemic. Studies have reported a significant increase in loneliness during the pandemic [17–19], with links to increases in depression, suicidality, and other mental health concerns [17, 18]. Restrictions on social activities have been associated with higher rates of loneliness [17] and measures such as quarantine, while reducing the spread of the illness, have been associated with a number of adverse psychological effects [20]. Authors of many studies on loneliness and COVID-19 have called for increased attention to addressing loneliness and related mental health concerns (e.g., [17, 18, 21]) that are unfolding along with the health crisis, while others have highlighted the additional stressors faced by grievers during pandemics and the down-stream consequences of burn out and traumatic stress [22].

Research has also explored social support in grief as salutary, as a mediator for proactive coping [23] and as a means to mitigate both the intensity and duration of psychological distress and poor physiological outcomes [3]. Some studies show that both the quantity and quality of social support may influence well-being for grievers [24]. Bereaved people who have more frequent contact with family and friends tend to report better quality of life, whether this support comes through technology (email and the internet) or in-person [25]. However, the number of contacts in a person's social network does not always predict well-being, pointing to the importance of the quality of relationships [26].

Meanwhile, other scholars note inconsistencies and limitations in the research supporting the notion that social support mitigates poor outcomes, particularly long term, in populations without what is commonly termed "complicated grief" [27]. In the only systematic review of its kind on traumatic grief and social support, the authors noted that current studies demonstrate "inconsistencies in quantitative conceptualisation of the measurement of social support" [28] and call for more qualitative research to provide "valuable insights to the bereavement experience in social settings" [28].

## What is social support?

There are four categories of social support defined in the literature that have been widely used over the past four decades. Structures and processes of social support, broadly speaking, include: 1) informational, 2) instrumental, 3) appraisal, and 4) emotional, all of which contribute to a sense of perceived and actual connection to a caring social system [29]. Intuitively,

there are needs related to loss that traverse all four categories of social support [30], such as funeral planning, financial aid, trauma and crisis care, and emotional experiences related to the loss. Different groups may benefit more from some types of support than from others, depending on their needs and circumstances [31]. Informational support may include logistical help on available services after death as well as advice, data, and information offered during a difficult or stressful time. Instrumental support is actionable aid that helps with specific tasks or provides necessary physical support such as food, shelter, transportation, and financial aid. Appraisal support is a more passive means of self-evaluation often enacted, for example, in peer-to-peer contact. It provides a means to assess one's self in a particular circumstance through like-others utilizing affirmation, feedback, and social equality [32, 33]. Finally, emotional support occurs through expressions of caring, compassion, trust building, and mutuality [32]. However, the conceptualization of these categories often lacks specificity in published studies [33, 34], perhaps especially in the scholarship of social support after trauma and grief.

### Social support and loneliness in grief

Pitman et al. [35] identified thwarted belongingness, withdrawal from others, lack of connection, and loneliness as significant risk factors for suicidality in grievers. The increased risk of suicide attempts for those experiencing grief was maintained even when controlling for cause of death and social stigma. Strong social support for those dealing with intense grief may not accelerate recovery [36], but it may improve the capacity to cope [37]. Typically, much of the responsibility for grief support falls on family and friends [38]. Yet, bereaved people often report insufficient support from these sources [39]. While each person has unique needs, those who have lost a spouse at an early age or a child often need more support than family or friends are typically able to give and may benefit the most from grief counseling or support groups [40]. Additionally, while the death of a loved one sometimes strengthens family relationships, often these relationships are strained, as family members may also be grieving, adding to the psychological distress of the bereaved; additionally, much of the support offered in early grief diminishes quickly [41] while the need for support continues.

Studies of the effects of grief support groups have reported mixed results [42], perhaps because few studies have examined specific behaviors in the group setting that are perceived as supportive. The same mixed results can be found in research around grief counseling [43], perhaps because what constitutes effective grief counseling is not well understood or defined in pedagogical models for providers [44]. Medical professionals, too, often have little training in dealing with the grief of their patients [45]. Overall, many providers who are present at the time of death are ill prepared to provide sufficient crisis or long-term support to bereaved individuals [46].

In sum, social support seems to help some bereaved people, particularly those with traumatic grief, that is, violent or sudden death of a close loved one or the death of a child, cope with psychological distress, while its absence may exacerbate poor physical and psychological outcomes [3]. Yet, a breakdown in social relationships after a loss is not uncommon [41], and loneliness- particularly salient during the COVID-19 pandemic- may exaggerate that effect for grievers [1], increasing the risk for poor outcomes. Despite findings that perceived social support mitigates risk in bereaved individuals [47], a dearth of qualitative research exists in understanding the specifics of this supportive care [28]. This study seeks to more specifically define good grief support and related actions and actors in this process.

### Method

Written consent was obtained and all data were collected anonymously following Institutional Review Board approval at Arizona State University. This cross sectional, qualitative descriptive

(QD) study explored ways in which grieving individuals interpret and define social support and its actions and actors after the death of a child, parent, partner, or sibling in adults over the age of 18.

QD is of particular utility in illuminating "poorly understood phenomenon" and commonly used in qualitative methods [48]. Qualitative description helps researchers construct particular meaning through respondents' descriptions of an event or experience [49, 50]. According to Tracy [51], excellent qualitative research should culminate in nuanced and rich data while meeting the following eight criteria: a worthy topic that is both relevant and timely, rich rigor, sincerity, credibility, resonance, significant contribution, ethics, and meaningful coherence. LaDonna, Taylor, and Lingard [52] note that "context, personal meaning, emotional and social nuances, and layers of detail" are what enrich our understanding of qualitative data (p. 348).

To that end, four open-ended, qualitative questions were developed in order to allow us to explore the actors and actions in social support after traumatic grief both within the context of acute crisis and in long term grief support: 1) What does social support mean to you? 2) What actions or responses felt supportive to you? 3) What actions or responses felt unsupportive to you? 4) How might others in your social support system do better to help you? These questions were carefully crafted to be simple, "focused and appropriate" in order to support "analytical procedures that offer robust insights into the social phenomena being explored" [52].

Practically, text boxes provided unlimited space for participants to expound on their subjective experiences, and these responses were a 'central focus' of meaning in the analyses [52]. Quantitative data, using a Likert scale, were also gathered to measure subjective satisfaction with both acute and long-term overall social support in the aftermath of traumatic grief. The survey utilized skip-logic, whereby participants were asked some questions only if they met criteria set by a specific previous question. The survey asked demographic and loss-related questions, Likert scale questions about how satisfied or dissatisfied they were with the quality of social support they received, and four open-ended questions specific social support following bereavement.

The survey was initially piloted with a small group to establish a level of validity and ensure that the questions were written in an appealing, engaging, and sensitive manner that provoked deep and rich responses. Cresswell and Miller's [53] criteria for determining validity in qualitative inquiry were used to inform the strategy, focusing particularly on incorporating opportunities for member-checking, investigator triangulation, and thick, rich description.

Utilizing snowball sampling through social media and several bereavement support organizations, a total of 372 grieving adults over the age of 18 years participated in a cross-sectional, anonymous, online survey asking them to define various aspects of social support during their bereavement experiences. Over the course of one week during January of 2020, data were collected online using Qualtrics following approval by the principal investigator's University Institutional Review Board.

Because of its methodological flexibility, content analysis (CA) was used to generate findings from the qualitative responses based on meaningful data that, through the process of recurrence and intentionality, creates cohesion and coherence [54]. The objective of CA is to present the 'big picture' that merges with the context, including the target population, their experience, and theory [55]. Qualitative responses were printed, read separately and in their entirety by two members of the research team, then unitized and contextualized [56] into a codebook. Through multiple, iterative-inductive rounds of data coding, open discussion, rated as effective as kappa values (as a means of managing and rationalizing any disagreements on which codes to apply) [57], intercoder consensus, and recoding, the findings were compared and validated in consultation with the literature [58].

All unitized data were also analyzed by the team to discern the degree of congruence with a priori codes in the existing theoretical literature on social support and categorized as: 1) emotional, 2) instrumental, 3) informational, and 4) appraisal. Of note is that emotional support as a construct is more difficult to quantify. For the purposes of these analyses, the working definition of emotional support used includes caring for another through the active expression of compassion and empathy, tenderness, validation of feelings, respect, listening, allowing emotions, and through the expression of loving kindness [59]. The research team paid careful attention to "any new questions or themes that emerge during the coding" [55]. For example, emotional support was distinguished from emotional acts of concern, which emerged as a separate theme in the analyses through investigator triangulation [60, 61], by identifying a tangible and direct action that held emotional meaning for participants.

## Results

### Descriptive

The survey sample ($n$ = 372) was predominantly female (91.4%), white (91.1%), partnered or married (69%), with college or graduate degrees (58.1%). The majority of respondents had experienced the death of a child (75.1%), followed by a spouse/partner (11.7%), a parent (7.9%), and a sibling (5.2%). The time since the loss was more than five years (43.3%), followed by one to three years (25.8%), three to five years (17.5%), and within the past year (13.4%). The most frequent cause of death was illness or disease (25.8%) followed by accident (19.2%), perinatal/infant death (12.6%), suicide (9.3%), unknown/undeterminable (7.9%), homicide (6%), and overdose (4.9%), with 14.3% choosing "other" as the cause of death.

### Social support

Respondents were asked to rate their perception of social support since their loved one's death on a scale ranging from excellent to very poor; 35.7% reported their overall support as excellent or good, 26.5% as adequate, and 37.9% as poor or very poor. Participants were asked to assess their degree of satisfaction with the initial social support they received from providers during the acute loss-related crisis, when applicable. Not all participants interacted with every category of provider. Response options ranged from extremely satisfied to extremely dissatisfied, shown in Table 1.

Mortuary staff received the largest percentage of ratings of effectiveness, with 65% reporting good or excellent satisfaction, followed by 63% for hospice staff, 55% for nurses, 47% for faith

**Table 1. Satisfaction with crisis support in grief.**

|  | Extremely Satisfied | Satisfied | Adequate | Dissatisfied | Extremely Dissatisfied | Mean | N |
|---|---|---|---|---|---|---|---|
|  | 1 | 2 | 3 | 4 | 5 |  |  |
| Mortuary staff | 47% (n = 141) | 18% (n = 55) | 22% (n = 67) | 6% (n = 19) | 7% (n = 20) | 2.08 | 302 |
| Hospice staff | 48% (n = 25) | 15% (n = 8) | 13% (n = 7) | 10% (n = 5) | 13% (n = 7) | 2.25 | 52 |
| Nurses | 38% (n = 85) | 17% (n = 37) | 14% (n = 32) | 11% (n = 25) | 20% (n = 44) | 2.58 | 223 |
| Faith leaders | 30% (n = 71) | 17% (n = 41) | 19% (n = 45) | 17% (n = 39) | 17% (n = 40) | 2.73 | 236 |
| Crisis response team | 32% (n = 25) | 10% (n = 8) | 17% (n = 13) | 17% (n = 13) | 24% (n = 19) | 2.91 | 78 |
| First responders | 32% (n = 45) | 9% (n = 12) | 23% (n = 33) | 16% (n = 22) | 21% (n = 29) | 2.84 | 141 |
| Law enforcement | 19% (n = 31) | 18% (n = 30) | 17% (n = 28) | 14% (n = 23) | 31% (n = 51) | 3.20 | 163 |
| Physicians | 21% (n = 51) | 14% (n = 34) | 22% (n = 54) | 14% (n = 34) | 30% (n = 72) | 3.17 | 245 |
| Hospital social workers | 22% (n = 32) | 13% (n = 19) | 17% (n = 25) | 14% (n = 20) | 34% (n = 49) | 3.17 | 145 |

**Table 2. Satisfaction with overall grief support.**

| | Extremely Satisfied | Satisfied | Adequate | Dissatisfied | Extremely Dissatisfied | Mean | N |
|---|---|---|---|---|---|---|---|
| | 1 | 2 | 3 | 4 | 5 | | |
| Pets/Animals | 78% (n = 194) | 11% (n = 27) | 10% (n = 25) | 1% (n = 2) | 0% (n = 0) | 1.55 | 248 |
| Online grief groups | 39% (n = 94) | 28% (n = 69) | 25% (n = 60) | 5% (n = 13) | 3% (n = 7) | 2.67 | 243 |
| Counselors/Therapists | 38% (n = 101) | 18% (n = 48) | 19% (n = 49) | 17% (n = 44) | 8% (n = 21) | 2.99 | 263 |
| In person grief groups | 34% (n = 63) | 24% (n = 46) | 19% (n = 36) | 15% (n = 29) | 7% (n = 14) | 3.65 | 188 |
| Friends | 25% (n = 86) | 27% (n = 92) | 22% (n = 74) | 16% (n = 54) | 9% (n = 32) | 3.31 | 338 |
| Family | 20% (n = 69) | 20% (n = 69) | 18% (n = 61) | 25% (n = 84) | 16% (n = 55) | 3.76 | 338 |
| Faith leaders | 17% (n = 34) | 22% (n = 44) | 19% (n = 38) | 21% (n = 41) | 20% (n = 39) | 3.86 | 196 |
| Colleagues | 12% (n = 30) | 19% (n = 46) | 32% (n = 78) | 20% (n = 49) | 16% (n = 38) | 3.95 | 241 |
| Neighbors/communities | 14% (n = 35) | 17% (n = 42) | 26% (n = 65) | 26% (n = 65) | 16% (n = 40) | 3.99 | 247 |

leaders, 42% and 41%, respectively, for crisis response and first responders, 37% for law enforcement, and 35% for physicians and hospital social workers.

Asked about satisfaction with social support generally, notably, of the 248 participants who had a relationship with their pets or other animals, 89% reported being extremely or mostly satisfied with the perception of support. Animals received the highest percentage of satisfaction among all categories of social support including support groups, counselors or therapists, friends, family, faith leaders, colleagues, and community members as shown in Table 2.

Extrapolating data specific to *dissatisfaction* with the degree of perceived social support, faith leaders, family, and communities/neighbors all either elicited *extremely or mostly dissatisfied* by 41–43% of the sample. This was followed by dissatisfaction with colleagues at 36%, counselors and friends each at about 25%, in-person support groups at 23%, and online support groups at 8.2%. The lowest rating of dissatisfaction was given to animals, at under 1%.

## Qualitative findings

**Survey results.** Respondents were invited to answer a number of open-ended questions in their own words.

*What does social support mean to you*? This question was answered by 91% (*n* = 338) of the sample. Informational support was mentioned only once. Instrumental support, in the form of practical aid, was mentioned by 5% (*n* = 16) of participants 17 times. Seven percent (*n* = 22) of participants specified appraisal support one time. Importantly, acts of emotional caring emerged in these data as a category of specific emotion-focused actions that traverse emotional and instrumental support, such as sending a card or reaching out through a phone call or text message, and were referenced 121 times by 23% (*n* = 78) of respondents. And, finally, emotional support yielded the highest percentage and frequency in responses with 64% (*n* = 215) of the sample mentioning it 292 times:

> "It means having a community of people who are safe to share your journey of grief with; who don't try to fix you or hurry you. . .people who let me say her name and tell stories about her."

> "It means people checking on me, inviting me to places, listening and remembering."

> "Having a community of people around us that are willing to listen and be there for each other."

*What actions or responses felt supportive to you*? This question elicited answers from 93% (*n* = 346) of the sample. Informational support was not mentioned, and appraisal support was

mentioned by 6% (*n* = 20) of the respondents 20 times. This took the form of connecting with people who had experienced similar grief in online or in-person support groups or with people in their community. The support received from like others included advice, understanding, and connection. While advice was generally not desired from sources of social support, advice and connection with "*like others*" was experienced as helpful:

> *"Support groups have been most helpful, knowing there were others who truly understand."*

> *"Advice from those who also lost a child has been helpful."*

Instrumental support logged 53 mentions by 13% (*n* = 44) of the sample and included assistance with childcare, food delivery, financial support, yard work, and housecleaning:

> *"Helping take care of my child. Helping with household chores."*

> *"Written notes, gifts, money for the expenses. . ."*

Emotional support was mentioned by 59% (*n* = 204) of respondents 316 times. Responses and actions related to "*being present*" and "*holding space*" were predominant and included descriptions like allowing the expression of grief, deep listening and "*quiet understanding*," availability and commitment of time with the griever, acceptance of emotional states, refraining from a "*need to fix*" or giving unsolicited advice, being actively open to grief, and a commitment to long term support by "*not fizzling out*:"

> *"Consistent communication, acceptance of my feelings, allowing me space as needed, listening without trying to fix or belittle my grief."*

> *"Just being present. Not trying to fix anything. Listening. Letting me talk about Thomas. Remembering Thomas. Honoring him."*

> *"Telling me that my grief is valid, that my feelings are real. Basically just allowing me to be."*

Sixty two percent of respondents (*n* = 214) mentioned acts of emotional caring 428 times:

> *"Being available week after week to walk and chat, gifts from neighbors of homemade cakes."*

> *"People checking in with me. People willing to simply listen and hold space for me and my pain. People who sent cards months and months even after. Bringing food. Buying gift cards for our family."*

> *"Checking in on me. Coming over to see how I am. Bringing dinner."*

The most frequently mentioned forms of emotional support included remembering the person who died, speaking their name, sharing memories of them, or acknowledging important dates related to them:

> *"ANYONE doing ANYTHING that lets me know they are thinking of him."*

> *"Just letting me mention his name without awkward silence or changing the subject."*

*What kinds of actions or responses felt unsupportive to you*? Eighty nine percent (*n* = 330) of respondents answered this question. No participant mentioned actions related to appraisal, informational, or instrumental support. Only two categories, emotional support (34%,

*n* = 112) and emotional acts of caring (63%, *n* = 208), were reported in these data for unsupportive responses or actions, in other words as examples of the failure to provide meaningful support in these areas. Specifically, respondents talked about the use of platitudes; judging or rushing grief; failure to approach or acknowledge loss; feeling abandoned by family, friends, and community members; avoidance of grief and griever; not listening; the perception that others were pretending the person who died had never existed; others' propensity to center their own needs and feelings above the primary griever; and offering unsolicited advice, especially about how to heal grief:

> *"Judging my grief, telling me they are tired of hearing it, facial expressions indicating irritation, annoyance. . . failing to acknowledge death days, birthdays, him at other holidays, indicating by words, silence, or actions that they are tired of my grief and tired of hearing about him or grief."*

> *"Never mentioning my daughter as if she never existed. . . friends stopped talking to us."*

> *"No one checked in on me, feeling alone and isolated."*

> *"Walking away from us when they see us. Changing the subject. Insisting I need to get out of the house because they miss the old me. Wanting me to participate in social events or visiting in the same way I did before Gabe was murdered. Not understanding I am forever changed."*

> *"I was surprised that the pastor who oversaw my son's funeral never checked on us."*

*How might others in your social support system do better to help you*? This was answered by 86% (*n* = 319) of the sample. Informational support was not mentioned. Appraisal support was mentioned by three participants one time (1%), mostly around more access to support groups, with one sibling noting "*there is no group*" available for this type of loss. Instrumental support in the form of providing food or help with chores was mentioned by 3% (*n* = 10) of the respondents 23 times:

> *"Do more practical help. Things like mow the lawn, watch the kids, do the dishes, bring meals."*

One-half of respondents indicated the desire for more emotional support 256 times (49%, *n* = 156). Participants talked about how important it is for others to listen, to learn about grief, to show up and not avoid them, and to accept their emotional state, to "*let me fall apart*." Overwhelmingly they wanted others to be present and to remember their loved one who died without the need to '*fix*' their grief and absent a timeline or agenda:

> *"If they would just allow me to express and honor what I feel."*

> *"Talk to me about him. SAY HIS NAME. He existed."*

> *"Just keep showing up for me and listen. Don't try to fix it."*

> *". . . not trying to fix what can't be fixed. By accepting the new sad, grieving me as someone who is ok to be with and not expecting me to feel like I need to get over this or move on and become the happy person I was. Not making me feel like I have to wear a mask and pretend I am someone I am not."*

> *"Remembering Michael with me and saying his name—telling me memories of him. I won't break if you talk about him, I want to honor and remember him forever."*

Acts of emotional caring were referenced by 48% of the sample (*n* = 154) 290 times, most often through active remembering (32%, *n* = 102) and reaching out to the grieving person (29%, *n* = 93):

> *"Reach out to me more often. Ask me more about my grief experience."*

> *"If they wait for me to reach out they are only pushing me further away."*

Five percent (*n* = 15) spontaneously wrote at least one time about ways animals support them emotionally:

> *"My dog was the person who was with me all the time. I think just having that soul there who can't say anything so it's like you know they're not saying the right or wrong thing, they're just there."*

> *"And sometimes when we walk, I talk to him about my son and he has no choice but to listen because he's a dog. . . a loving comforting animal."*

> *"They brought my dog to the hospital to see me. I think that's the thing that probably was the divider between going into depression and not."*

## Discussion

Findings highlight that many grievers experience dissatisfaction with social support from a variety of actors despite the increased need for support in bereavement and the high risk for poor emotional, mental, and physical outcomes. Just over one-third of the sample rated the level of social support they received post-loss as excellent or good, with 38% rating it as poor or very poor. Specific actors who provided good or excellent social support in more than 50% of the sample included animals, support groups, counselors or therapists, and friends, while the percentage was below 50% for family, faith leaders, colleagues, and community members.

These data also illuminate the primacy of emotional support and acts of emotional concern as shown in Fig 1. Specifically, emotional support in grief can be defined as being present and holding space for the griever, more specifically distilled to listening and allowing grief without judgment, platitudes, or an agenda. This means being available for the griever, spending time with the griever, centering the griever's needs, and not imposing a time limit on grief. Acts of emotional concern can be expressed by remembering (asking about, sharing memories of, and speaking the name of) the person who died, acknowledging important dates, and engaging in emotionally sensitive communication. It is not simply the amount of contact with family and friends that equates to good social support, but also the quality of the connection, with the presence of emotional support and expression of caring being highly valued.

One participant distilled it down to the feeling of being loved:

> *"I feel supported by my dogs. Simply because the love is unconditional."*

These findings echo previous reports underscoring the importance of emotional support for grieving individuals [59, 62–64]. Concrete examples of valued emotional support and acts of concern in the current study included reaching out to contact the bereaved person, actively allowing the expression of grief and acceptance of emotional states, sensitive and compassionate communication, and actively remembering the person who died. Also noted was the importance of support not just in the immediate aftermath of the loss but over a longer period of time.

## Supportive Actions in Grief

**ACTS OF EMOTIONAL CARING**
Remembering the person who died
Speaking his/her name
Sharing memories
Acknowledging important dates and holidays
Emotionally sensitive communication

**EMOTIONAL SUPPORT**
Being present
Allowing the expression of grief
Listening
Being available
Devoting time
Being open to grief emotions
Not trying to fix or rush grief
Refraining from unsolicited advice
Timeless support

**INSTRUMENTAL SUPPORT**
Assistance with childcare, meals, financial support,
yard work, and housekeeping
Written notes and gifts

**APPRAISAL SUPPORT**
Sense of connection with like others
Support groups
Extended communities (faith, schools, neighborhoods)

**Fig 1. Supportive actions in grief.**

Less salient in these data was the interpretation of instrumental, appraisal, and informational support as good social support. Instrumental support was effective when expressed through helping with meals, childcare, housekeeping, and written notes and gifts. One important aspect of instrumental support deserving of attention may be the classic mistake of saying, "...*call if you need anything,*" without any follow-up. Participants appreciated others actively reaching out to them to offer practical aid. Appraisal support meant connecting with like others through grief support groups, in person and online, and on social media. Time spent with others, both online and in person, who share a common tragedy of loss was reported as supportive in these data. Informational support was largely absent, echoing previous findings suggesting informational support is not among the most helpful forms of social support in bereavement [63]. And, while some respondents noted that they appreciated informational support and guidance through their support groups, others perceived unsolicited information and advice to be unhelpful. Also, the salutary effect of informational support may be more specific to certain subgroups of the bereaved. For instance, in one study with widowed fathers of dependent children, participants expressed a desire for more information and guidance than was offered [65].

Perhaps most interesting is the finding that the overwhelming majority of respondents rated animals as highly satisfactory. Animals are not usually included in conceptualizations of possible sources of social support in bereavement. However, research in other fields has provided evidence of the beneficial impact of interactions with animals on human health and well-being [66, 67]. The recognition of animals as sources of meaningful social support in bereavement may open up avenues for more effective support to grievers, especially in socially isolated populations and under certain conditions such as quarantine or physical distancing.

Also instructive are findings about the lowest rated groups, amongst which hospital social workers, followed by physicians and law enforcement, ranked highest in dissatisfaction during acute care, and faith leaders ranked third to lowest in overall satisfaction with grief support. Specifically, these provider groups often encounter others in crisis and, with the exception of law enforcement, are in the caring professions. Emotional support is essential for vulnerable populations [62], which are often served by social workers, hospital staff, and faith leaders. That there is such low satisfaction with the provision of support at a time of such great need is worthy of further study. There may be several reasons for this. First, the "prevailing views of emotions as the antithesis of reason" in pedagogical models [62] requires a serious reappraisal [68, 69]. Scholars have noted the lack of attention to skillfully working with emotions and developing emotional competence in medical education and social work education classrooms [69, 70]. For example, content on emotions in social work pedagogy tends to focus on the compartmentalization of emotions rather than learning to express, process, and integrate emotions [71]. Even in field education, supervisors may refer to students' emotional responses as "signs of immaturity" stemming from "unresolved personal problems" [72], absent evidence of such assertions. In particular, social work historically serves the most vulnerable populations [68], many of whom are experiencing trauma and grief. They also serve many frequent users of healthcare systems who are at highest risk for poor social determinants of health [68, 73]. Additionally, hospital social work roles have largely shifted from supportive counseling to discharge planning [74], particularly in bereavement, and are relegated to the margins, perhaps because they are comparatively low-cost service providers in the medical setting.

The essentiality of emotion-focused support in bereavement also warrants further attention. While emotional support is perhaps most valued in bereavement and can decrease psychological distress [3], social networks are not always capable of providing adequate emotional support. In addition, the current findings echo previous findings (e.g., [65]) reporting a mismatch between the kinds of social support offered and what grievers desire. These data highlight the

importance of educating those in bereaved individuals' social networks to be more responsive to the emotional needs of grievers and for pedagogical and practice models to promote and teach emotionally supportive behaviors.

While this study adds to the understanding of social support in bereavement, it has a number of limitations. Self-reports are subject to recall bias and other shortfalls occurring during this type of cross-sectional data acquisition [75]. The sample was demographically homogenous, predominantly female, and most respondents experienced the death of a child, or other traumatic death such as the suicide of a parent, and findings may not be representative of relatively uncomplicated grief in the general population. These data, thus, seem to support Stroebe and colleagues [27] finding that social support may be more essential when grief is traumatic and complicated. There may also be differences in the necessary social support structures and processes based on gender, something that should be more specifically investigated in future studies, relationship to the deceased, manner of death, or other factors, or differences in actors and actions depending on the same variables. For example, men and women may desire and elicit different types of social support while the death of a parent to suicide may elicit more social avoidance than the death of a parent to cancer, leaving the bereaved with more unmet social support needs. Additionally, future research on effective bereavement social support could specifically focus on age groups to discern any differences or similarities across the lifespan. Since many of the studies on social support in bereavement focus on uncomplicated bereavement, future studies could explore emotional and physical health outcomes as they relate to satisfaction with traumatic grief support longitudinally. Despite the aforementioned limitations, this study adds to the body of scientific data about social support in bereavement, offering insights into participants' lived experiences.

## Conclusion

The burden of loneliness in grief, especially when traumatic, is costly and presents a potential public health crisis, particularly in the aftermath of the COVID-19 pandemic [1, 17, 18, 21]. Findings from this study can be used to help educate others about the needs of grievers and how to adequately support them to reduce some of the cost to families and society. Improvements in social support that focus on emotions, in turn, might decrease the common loneliness experienced by grievers and thus the associated health risks, both during and beyond the scope of the COVID-19 pandemic. However, pedagogical models, often "woefully deficient in educating its learners about the role of emotions in healthcare," emphasize, instead, emotional distancing and detachment [69]. Not only is this the case in medicine but also in social work practice [62, 69]. Faculty could restructure educational objectives to emphasize the necessity of tending to the emotions of patients and clients in addition to the importance of personal growth toward emotional intelligence. Future research could more precisely explore the effects of emotional support, specifically, in mitigating poor psychological and health outcomes in the bereaved. Further, community-based education programs might focus on emotional health, in addition to mental health, with a turn toward coping with painful emotions and helping support others, in faith and school systems, in communities, and at work.

A particularly interesting finding is the high level of satisfaction reported with animals as sources of social support. Animals may be an especially important source of emotional support during conditions involving social isolation, such as the COVID-19 pandemic when contact with other people is limited, or during experiential conditions such as the loneliness so common in bereavement. Further research could investigate the ways in which animals are perceived as beneficial in grief more thoroughly, but the adoption of pets could be one avenue by

which to promote well-being and reduce loneliness during the pandemic, especially for those who are not able to access strong social support networks.

When it comes to good grief support, perhaps we may have much to learn from our fellow non-human animals.

## Supporting information

**S1 Questionnaire.**
(DOCX)

**S1 Data.**
(XLSX)

## Author Contributions

**Conceptualization:** Joanne Cacciatore, Lori Barnes Jackson.

**Data curation:** Joanne Cacciatore, Lori Barnes Jackson.

**Formal analysis:** Joanne Cacciatore, Kara Thieleman, Lori Barnes Jackson.

**Investigation:** Joanne Cacciatore.

**Methodology:** Joanne Cacciatore, Lori Barnes Jackson.

**Project administration:** Lori Barnes Jackson.

**Supervision:** Joanne Cacciatore.

**Validation:** Joanne Cacciatore, Ruth Fretts.

**Writing – original draft:** Joanne Cacciatore.

**Writing – review & editing:** Joanne Cacciatore, Kara Thieleman, Ruth Fretts, Lori Barnes Jackson.

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
