## [Decision Letter · Decision Letter 0]

18 Mar 2021

PONE-D-21-04230

Good Grief Support:

Exploring the Actors and Actions in Social Support

after Traumatic Grief

PLOS ONE

Dear Dr. Cacciatore,

Thank you for submitting your manuscript to PLOS ONE. After careful consideration, we feel that it has merit but does not fully meet PLOS ONE’s publication criteria as it currently stands. Therefore, we invite you to submit a revised version of the manuscript that addresses the points raised during the review process.

Reviewers have found that the manuscript is promising. It is important that you include more and clear information on the method section of the manuscript. Please, follow Reviewer 1 suggestions regarding the missing information about the research design and the data analysis. Before submitting a revised version of the manuscript please adhere to the COREQ guidelines for reporting qualitative research (https://www.equator-network.org/reporting-guidelines/coreq/) that may help you to describe the methodology of the study.

We look forward to receiving your revised manuscript.

Kind regards,

Manuel Fernández-Alcántara, Ph.D.

Academic Editor

PLOS ONE

Journal Requirements:

Reviewers' comments:

Reviewer's Responses to Questions

**Comments to the Author**

1. Is the manuscript technically sound, and do the data support the conclusions?

Reviewer #1: No

Reviewer #2: Yes

Reviewer #3: Yes

2. Has the statistical analysis been performed appropriately and rigorously? 

Reviewer #1: N/A

Reviewer #2: Yes

Reviewer #3: Yes

3. Have the authors made all data underlying the findings in their manuscript fully available?

Reviewer #1: No

Reviewer #2: Yes

Reviewer #3: No

4. Is the manuscript presented in an intelligible fashion and written in standard English?

Reviewer #1: No

Reviewer #2: Yes

Reviewer #3: Yes

5. Review Comments to the Author

Reviewer #1: The article “Good Grief Support: Exploring the Actors and Actions in Social Support after Traumatic Grief”, assuming that social support enhances wellbeing and health especially in traumatic grief, consider how mourners interpret and define social support. In the scenario of the current pandemic, the study assessed mourners satisfaction of social support in traumatic grief, using four categories of social support as a framework. Findings suggest inadequate satisfaction from professional, familial, and community support. Pets emerged with the most satisfactory ratings. Further, findings suggest that emotional support is the most desired type of support following traumatic loss.

The introduction is too long and perhaps too detailed with respect to the concept of social support. This part can be summarized. Less clear, however, is the description of the effects of Self Help Groups, one of the most important forms of social support offered to traumatic grief (lines 113-120). I highlight this work that may be helpful: “Testoni, I., Francescon, E., De Leo, D., Santini, A., & Zamperini, A. (2019). Forgiveness and Blame Among Sui-cide Survivors: A Qualitative Analysis on Reports of 4-Years Self-Help-Group Meetings. Community Mental Health Journal, 55(2), 360-368. doi:10.1007/s10597-018-0291-3”

The methodology part is very messy. It is best to follow this structure: a) clearly define the objectives; b) clearly define the participants and describe all demographic characteristics of the participants; c) clearly describe the research design and justify the methodologies; d) clearly describe the steps of the analyses. The qualitative approach used is unclear. Direct quotes in the methodological section make the authors seem inexperienced in the field.

The relationship between the research and the pandemic is not at all clear: when was the survey done? were clear references to the pandemic made? was it clear to participants that the survey was about or could be influenced by the pandemic experience?

The topic is very interesting but the presentation of results at the moment is still very haphazard.

Reviewer #2: Thank you for asking me to review this interesting paper. Here are my impressions that I hope are helpful and constructive.

In the abstract, the authors summarize the research objective and the main findings. However, a mention of the methodology used is missed.

The introduction section identifies comprehensively the topic and explains how the study relates to this previously published research. The need for this study is justified.

The tables provided as supporting information are clear and legible and support the findings, however they lack titles.

Regarding the methodology, it provides information on how the research was carried out. One minor comment: Line 160 mentions triangulation as a validity strategy. Detailing what type of triangulation would be clarifying.

In the results and discussion, two elements caught my attention: The disparity of etiologies of death and a markedly female sample. Although both are mentioned in the study limitations, and especially the cause of death is justified with evidence, gender is only mentioned in line 418. It would be interesting to address this aspect, and even show if the perception of social support differs whether the death is unexpected or traumatic or it is not.

Finally, the text lacks a section on conclusions.

Reviewer #3: Thank you for the opportunity to review manuscript ID# PONE-D-21-04230, “Good Grief Support: Exploring the Actors and Actions in Social Support after Traumatic Grief.” This paper presents the results of a mixed method study assessing bereaved individuals’ satisfaction with social support in the aftermath of a traumatic loss. Strengths of this paper include the large sample size and discussion of specific behavioral examples of social support identified via qualitative methods. I commend the authors for this important work, which has implications for providers working across a variety of different disciplines, and have only a few suggestions for the authors’ consideration.

1) In the beginning of the Method section, the authors note that the data presented here was part of a larger effort to define perceptions of specific actions and actors of effective post-bereavement social support. To clarify, was this data collected as part of a larger study, and if so, is this sample a subset of the sample from the larger study? Additional clarification would be helpful.

2) When describing the study methodology, please discuss any explicit inclusion/exclusion criteria.

3) When presenting descriptive statistics, I noticed no information on age was provided. Did the authors collect any information on age, and, if not, is there any reason to think that this might be an important area of future inquiry worth mentioning in the Discussion section (e.g., ensuring that the kinds of social support mentioned are similarly relevant across the lifespan)?

4) On page 10, lines 196-197, the authors mention that participants were asked to assess their degree of satisfaction with the initial social support they received from various individuals during the acute loss-related period. I recommend mentioning this in the Method section when discussing survey content, along with a description of the various relationship groups/categories included in this survey. Additionally, how was the acute loss-related period defined on the survey?

5) On page 19 of the Discussion section, the authors suggest several interesting and important reasons why provider groups such as hospital social workers, physicians, law enforcement, and faith leaders are among the lowest rated groups in terms of satisfaction with support. In addition to issues with training models and curricula, could occupational burnout also be relevant here given the intense demands place on these folks with often minimal organizational support? I might mention the literature on burnout here as well just for context.

6. PLOS authors have the option to publish the peer review history of their article (what does this mean?). If published, this will include your full peer review and any attached files.

Reviewer #1: No

Reviewer #2: No

Reviewer #3: **Yes: **Joah Williams

---

## [Author Response · Author response to Decision Letter 0]

13 Apr 2021

Thank you so much for improving our manuscript. We are included a list of changes as recommended by reviewers and responses. We have uploaded an anonymized database. We have included a track changes version and clean version of the manuscript. We have followed the suggestions for methods of R1 and relevant COREQ guidelines. We hope these changes are satisfactory. Thank you!

---

## [Decision Letter · Decision Letter 1]

14 May 2021

What is Good Grief Support?

Exploring the actors and actions in social support after traumatic grief

PONE-D-21-04230R1

Dear Dr. Cacciatore,

We’re pleased to inform you that your manuscript has been judged scientifically suitable for publication and will be formally accepted for publication once it meets all outstanding technical requirements.

Kind regards,

Manuel Fernández-Alcántara, Ph.D.

Academic Editor

PLOS ONE

Additional Editor Comments (optional):

Reviewers' comments:

Reviewer's Responses to Questions

**Comments to the Author**

1. If the authors have adequately addressed your comments raised in a previous round of review and you feel that this manuscript is now acceptable for publication, you may indicate that here to bypass the “Comments to the Author” section, enter your conflict of interest statement in the “Confidential to Editor” section, and submit your "Accept" recommendation.

Reviewer #1: All comments have been addressed

Reviewer #2: All comments have been addressed

2. Is the manuscript technically sound, and do the data support the conclusions?

Reviewer #1: Yes

Reviewer #2: Yes

3. Has the statistical analysis been performed appropriately and rigorously? 

Reviewer #1: Yes

Reviewer #2: Yes

4. Have the authors made all data underlying the findings in their manuscript fully available?

Reviewer #1: Yes

Reviewer #2: Yes

5. Is the manuscript presented in an intelligible fashion and written in standard English?

Reviewer #1: Yes

Reviewer #2: Yes

6. Review Comments to the Author

Reviewer #1: The articile is quite interesting and wel written.

After the revision it can be published.

Congratulation on this result.

Reviewer #2: (No Response)

7. PLOS authors have the option to publish the peer review history of their article (what does this mean?). If published, this will include your full peer review and any attached files.

Reviewer #1: No

Reviewer #2: No

---

## [Editor Report · Acceptance letter]

19 May 2021

PONE-D-21-04230R1 

What is Good Grief Support? Exploring the actors and actions in social support after traumatic grief 

Dear Dr. Cacciatore:

I'm pleased to inform you that your manuscript has been deemed suitable for publication in PLOS ONE. Congratulations! Your manuscript is now with our production department. 

Kind regards, 

on behalf of

Dr. Manuel Fernández-Alcántara 

Academic Editor

PLOS ONE